# Biomarkers of Deoxynivalenol Toxicity in Chickens with Special Emphasis on Metabolic and Welfare Parameters

**DOI:** 10.3390/toxins13030217

**Published:** 2021-03-17

**Authors:** Insaf Riahi, Anna Maria Pérez-Vendrell, Antonio J. Ramos, Joaquim Brufau, Enric Esteve-Garcia, Julie Schulthess, Virginie Marquis

**Affiliations:** 1Animal Nutrition Department, Institute of Agrifood Research and Technology (IRTA Mas Bové), 43120 Constanti, Spain; anna.perez@irta.cat (A.M.P.-V.); joaquim.brufau@irta.cat (J.B.); enric.esteve@irta.cat (E.E.-G.); 2Applied Mycology Unit, Food Technology Department, University of Lleida, UTPV-XaRTA, Agrotecnio, Av. Rovira Roure 191, 25198 Lleida, Spain; antonio.ramos@udl.cat; 3Phileo by Lesaffre, 137 Rue Gabriel Péri, 59700 Marcq en Baroeul, France; j.schulthess@phileo.lesaffre.com (J.S.); v.marquis@phileo.lesaffre.com (V.M.)

**Keywords:** deoxynivalenol, chickens, deoxynivalenol-3 sulphate, biomarkers

## Abstract

Deoxynivalenol (DON), a trichothecene mycotoxin produced by *Fusarium* species, is the most widespread mycotoxin in poultry feed worldwide. Long term-exposure from low to moderate DON concentrations can produce alteration in growth performance and impairment of the health status of birds. To evaluate the efficacy of mycotoxin-detoxifying agent alleviating the toxic effects of DON, the most relevant biomarkers of toxicity of DON in chickens should be firstly determined. The specific biomarker of exposure of DON in chickens is DON-3 sulphate found in different biological matrices (plasma and excreta). Regarding the nonspecific biomarkers called also biomarkers of effect, the most relevant ones are the impairment of the productive parameters, the intestinal morphology (reduction of villus height) and the enlargement of the gizzard. Moreover, the biomarkers of effect related to physiology (decrease of blood proteins, triglycerides, hemoglobin, erythrocytes, and lymphocytes and the increase of alanine transaminase (ALT)), immunity (response to common vaccines and release of some proinflammatory cytokines) and welfare status of the birds (such as the increase of Thiobarbituric acid reactive substances (TBARS) and the stress index), has been reported. This review highlights the available information regarding both types of biomarkers of DON toxicity in chickens.

## 1. Introduction

The consumption of poultry meat has shown an increase from 2016 (116,845.36 thousand metric ton) to 2018 (120,884.63 thousand metric ton) [1]. Moreover, during the period 2020–2029, global livestock production is expected to expand by 14%, poultry remaining the fastest growing meat accounting for about half of the projected increase in total meat output, while world population is expected to grow only by ca 1% per year [1].

The goal of the poultry industry is to ensure high productivity without jeopardizing product quality and safety. For this reason, ensuring microbiological and toxicological safety of poultry feed entails a major challenge.

Mycotoxins are fungal metabolites frequently found in feeds that can compromise the health of animals and humans [2]. Besides, mycotoxin feed contamination leads to important economic losses in animal production [3]. Mycotoxins can cause a large range of diseases, as well as death, to both humans and animals [4]. The most important mycotoxins associated with poultry health and productivity problems are aflatoxins (AF), fumonisins (FBs), ochratoxin A (OTA), zearalenone (ZEN) and trichothecenes such as deoxynivalenol (DON), and T-2 toxin [5].

As DON is the most frequently encountered mycotoxin in cereal commodities [6], it is not surprising that DON is a major contaminant of poultry feed. A recent large scale survey monitored the occurrence of mycotoxin from 2008 to 2017 in 74,821 samples of finished feed, maize, maize dried distillers grains with soluble (DDGS), maize silage, soybean grains, soybean meal, wheat, barley, and rice collected from 100 countries for the presence of AFB1, DON, ZEN, FBs, OTA, and T-2 [7]. DON was the most prevalent *Fusarium* mycotoxin followed by FBs and ZEN, mycotoxins were detected in 64%, 60%, and 45% of all samples, respectively [7].

DON is produced predominantly by *Fusarium graminearum* and *Fusarium culmorum* and, at to a lesser extent, by *Fusarium cerealis* and *Fusarium pseudograminearum*, under conditions of high moisture and low temperature [8]. DON mainly occurs when grains are still in the field, although it can also occur during improper storage [9]. As all trichothecenes, DON contains a tetracyclic sesquiterpenoid 12,13-epoxytrichothecene core, responsible for their cytotoxicity, and a 9, 10 double bonds with various side chain substitutions [10]. Structurally, DON is a polar compound composed of three free hydroxyl groups (-OH) which participate also in its toxicity [11]. DON is a thermostable compound and also is able to resist low pH levels [12]. DON has been classified by the International Agency for Research on Cancer (IARC) in group 3 as noncarcinogenic to humans [13]. At cellular level, DON is a protein synthesis inhibitor, which induces ribotoxic and oxidative stress [14,15,16]. Acute DON mycotoxicosis in broiler chickens is rare to occur due to the extremely high feed contamination [8]. Only one study reported the Acute DON mycotoxicosis in broiler chickens which is characterized by extensive ecchymotic hemorrhages, deposition of urates, alteration of the nervous system, and inflammation of the upper gastrointestinal tract [17]. However, the most common mycotoxicosis is due to chronic exposure from low to moderate level of DON and resulted in alteration of the production and health status of the birds which is manifested in alterations of performance, immune, intestinal, physiological, and welfare parameters [18,19,20].

Susceptibility to DON exposure depends on the affected species [21]. Poultry, for example, are considered resistant to DON mycotoxicosis in comparison with pigs, due to differences on the absorption, distribution, metabolism, and excretion (ADME) process of each species [22]. The guidance level of EU in poultry feed is about 5 mg/kg while for pigs is only 0.9 mg/kg [23].

The presence of mycotoxins in food and feed is practically unavoidable due to their stability. Moreover, the simultaneous presence of different mycotoxins in the recent survey is the rule rather than the exception [7]. The simultaneous presence of DON and ZEN, or DON and FBs were the most observed (48%) in the case of finished feed. It happens in a similar way with corn, with co-occurrences of DON and ZEN in 39%, and DON and FBs in 49% of the samples. The most frequent combination in wheat was DON and ZEN, which was detected in 28% in the samples [7].

Therefore, in order to minimize the risks posed by mycotoxins, the use of feed additives known as mycotoxins detoxifiers seems to be a good strategy to counteract the negative effects induced by mycotoxin, and this is the most common practice used today [24]. In 2019, the feed mycotoxin detoxifiers market was valued at USD 1204.0 million and is expected to grow at a Compound Annual Growth Rate (CAGR) of 6.4% during the forecast period (2020–2025) [1]. North America and Asia Pacific are the greatest zone using detoxifiers. The market in Asia Pacific is projected to be the fastest-growing, due to large livestock population and their increasing growth rate [25].

The efficacy and safety of these feed additives has been proven through the evaluation of biological biomarkers related to mycotoxin toxicity in a target species [26]. The use of biological biomarkers is highly reliable because they provide a basis for the designing of the in vitro and in vivo trials [26]. In this context, research has been conducted to demonstrate the efficacy of detoxifying-agents on unspecific biomarkers such as zootechnical parameters, organ weights, and physiological, immunological, and welfare indicators [27]. However, in the scientific opinion focusing on the experimental design of in vivo studies on the efficacy and safety testing of mycotoxin-detoxifying agents, the European Food Safety Authority (EFSA) has listed several relevant end-points for the different mycotoxins [28]. These end-points, also named biomarkers of exposure, are those directly related to mycotoxin toxicity. For DON, the most relevant end-points proposed are DON itself and its metabolite deepoxy-deoxynivalenol (DOM-1) in blood [28].

This review highlights the biomarkers of exposure and biomarkers of effect of DON in chickens, with special emphasis on metabolic and welfare parameters, in order to identify its impact and therefore facilitate the evaluation of the efficacy of detoxifying-agents. This review may also represent a useful tool for diagnosis of birds in the field.

## 2. Biomarkers of Exposure 

Specific well-characterized biomarkers have shown to predict relevant clinical outcomes across a variety of treatments and populations [26]. Up to date, the use of biomarkers has become commonplace, and biomarker-driven research has been proposed as a successful method [29]. The biomarkers of exposure are defined as the metabolites estimated in biological fluids upon the exposure to xenobiotics of individuals [29]. The biomarkers of exposure must be specific for each mycotoxin and target species, and the analytical method used for its detection must be validated for each biological matrix considered [28]. As previously mentioned, EFSA has indicated that the relevant biomarkers directly related to exposure of DON are DON and DOM-1, revealing that de-epoxidation is the most important pathway of metabolization of DON [28]. However, research conducted in different species revealed that the metabolization pathway of DON is species-dependent, and de-epoxidation has shown not to be an important pathway of metabolization for poultry species [30].

The metabolization of DON may be changed if the birds are exposed to DON in a chronic feeding design or to intravenous injection or oral bolus of the synthetic or labeled DON, also by the biological matrices and the sensitivity of the method of analysis used. The most relevant biomarkers of DON exposure in chickens and their methods of determination are reported in Table 1.

After chronic feeding of DON at 7.54 or 9.5 mg/kg to broiler chickens, DON was quantified as the main metabolite in plasma using LC–MS/MS with limit of quantification (LOQ) (0.1 to 1.25 ng/mL) [32,34]. However, the concentration of DON in plasma was under the LOQ (1.25 or 23.3 ng/mL) when broilers were fed lower (1 or 2.44 mg DON/kg feed) or equal to the guidance level of DON in poultry feed (5 mg/kg feed) [31,33,34].

On the other hand, DON was the only metabolite detected in the plasma when broilers were exposed to the equivalent to guidance level (approximately 0.5–0.75 mg/kg BW) intravenously or by oral gavage in a two-way cross-over design, using LC-MS/MS with LOQ (0.1–2.5 ng/mL) [22,38,39]. Furthermore, after 2 h post-administration no DON levels in broilers plasma above LOQ were detected [38]. The study of toxicokinetic parameters upon quantification of DON in plasma revealed a low absolute oral bioavailability (19.3%) or absorbed fraction (10.6%) due to the poor absorption of DON [22,38,39]. This low absorption may be partially related to the rapid transit time in the gastrointestinal tract of birds. The authors suggested also that this poor bioavailability could be associated to the metabolization of DON by the high bacterial content in the gastrointestinal tract of chickens before the main site of absorption [22]. Additionally, it has been reported that the high clearance with the rapid elimination half-life might be the reasons why poultry are relatively tolerant to DON mycotoxicosis [38].

Awad et al. [33] indicated that after exposure of DON in feed (1 or 5 mg/kg) to broiler chickens for 35 days, DON concentration was recovered around 10–12% in gizzard, 18–22% in cecum and only 6% in excreta of the DON administered. DON was not detected in liver and bile. Moreover, the DON concentration in the content (digesta) of gizzard, cecum, rectum, and excreta increased in a dose dependent manner. In contrast, DON could not be quantified in bile, liver, breast meat, and kidney [31,34]. The analysis of DON and its metabolites using HPLC in broiler chickens receiving an oral administration of tritium-labelled DON at the dose of 2.5 mg/kg BW for 5 days, indicated that DON was transiently distributed and rapidly eliminated in all tissues [35]. Low DON concentrations were detected in kidney, liver, heart, lung, spleen, and brain and the higher level was detected in small intestine 6 h post-administration.

With regard to DOM-1, several studies analyzed this metabolite in plasma, in different organs, or in excreta after chronic DON feeding, intravenous injection or oral gavage using LC–MS/MS [22,30,31,32,33,34,36,38,39,40]. However, only in very few studies DOM-1 could be quantified [32,34].

All studies mentioned followed the guidelines proposed by EFSA and quantified DON and DOM-1 as possible metabolites of exposure of DON in different biological matrices of chickens, but they found inconsistent results. The inconsistent results may be partially explained by different levels of 3 and 15 acetyl-DON in the feed. The lack of detection of DOM-1 could be explained by the fact that sulfation occurred so quickly that only DON-3S can be transformed to DOM-3S.

The high-resolution mass spectrometry (HRMS) is a very recent analytical method used to identify or detect the untargeted or novel metabolites, biotransformation products and/or modified fungal metabolites [41]. Since 2014, and with the improvement of the analytical methods it was reported that the most abundant metabolite in case of broiler chickens after receiving DON mycotoxin was a phase II metabolite named DON-3 sulphate (DON-3S) [35]. Still, the formation of DON-3S was observed also in turkeys, pigeons and laying hens [30,37,42,43]. Furthermore, in poultry no DON glucuronidation has been observed [30].

DON-3S was identified the first time as a new and major metabolite in poultry in the study of Wan et al. [35]. Among the three new metabolites identified (DON-3S, 10-DON-sulfonate, and 10-DOM-1-sulfonate), DON-3S was the major metabolite found in excreta of chickens (88.6%) after the oral administration of tritium-labeled DON at the dose of 2.5 mg/kg BW for 5 days. [35].

Subsequently, Devreese et al. [30], using HRMS analysis of plasma of turkeys and broiler chickens, found that DON-3S was the major metabolite after oral gavage as well as after intravenous injection of 0.75 mg DON/kg BW [30]. Broiler chickens biotransform or metabolize DON more extensively to DON-3S than turkeys [30]. The authors suggested that sulfation mainly occurred in the enterocytes of the intestinal mucosa [36]. Similarly, the analysis by LC–HRMS indicated that DON-3S was mainly present in plasma of broiler chickens dosed at 0.5 mg DON/kg BW after intravenous or oral DON administration, respectively; and that DON-3G was not hydrolyzed to DON in this species [40]. The authors concluded that chickens are less sensitive to DON toxicity and explained this relative tolerance by the rapid gastrointestinal transit time or by the extensive and rapid sulfation that occurred.

Schwartz-Zimmermann et al. [36] demonstrated that DON-3S was the best biomarker of exposure in all poultry species. In chickens, for example, biological recovery of DON-3S in excreta reached 80% after administration of DON (1.7 mg/kg feed). These authors indicated that DON is rapidly absorbed from the gastrointestinal tract between crop and jejunum. Then, they suggested that after absorption, DON is extensively biotransformed to DON-3S in the intestinal mucosa, liver and/or kidney. The elimination of DON-3S is rapid and complete into the cloaca via urine by the kidney (via the renal portal vein system in birds), or might be in part be subject to biliary excretion and excreted via feces, or back into the gastrointestinal tract via bile by the liver [36,43].

It was demonstrated also that the most suitable biomarker for exposure of DON in poultry is DON-3S detected in different biological matrices (plasma and excreta); after a single intra-crop bolus administration of 0.5 mg DON/kg BW to broiler chickens aged 21 days and weighed 1 kg, DON-3S was detected and DON was not found. The maximum peak area of DON-3S was achieved in plasma after 30 min post administration and after 3–6 h in dried excreta [37].

In conclusion, DON and DOM-1 are not considered ideal biomarkers in the different biological matrices of chickens. Interestingly, the most abundant metabolite in plasma and excreta of chickens is DON-3S which could be considered the most suitable biomarker of exposure of DON in chickens. This metabolization is a detoxification pathway [44]. On the other hand, the toxicokinetic behavior of DON in chickens was characterized by low absorption, rapid gastrointestinal transit time, transient distribution, high clearance, and rapid elimination. All these factors may explain the low susceptibility of broilers to DON toxicity.

## 3. Biomarkers of Effect

A biomarker of effect is evaluated through a biochemical, physiological, immunological, behavioral, or other organism’s alteration [29]. Depending upon the magnitude, the biomarker of effect can be recognized as associated with an established or possible health impairment or disease [29]. Biomarkers of effect are biomarkers indirectly related to mycotoxin toxicity. The biomarkers of effect of DON toxicity in chickens highlighted in this review are productive parameters, organ weights, morphology of small intestine, biochemical and hematological parameters, biomarkers related to immune system (common vaccines response as a part of humoral immune response, cellular immune response, and production of proinflammatory cytokines), and biomarkers related to chicken welfare. To our knowledge, this is the first time that the indicators of toxicity in chickens related to welfare are grouped in a review.

### 3.1. Productive Parameters 

Studies about the effects of DON on productive parameters in chickens are listed in Table 2.

The chronic ingestion of DON can cause decrease of feed consumption, body weight gain (BWG) and feed efficiency in chickens [62]. Even at lower concentrations than the recommended guidance value (5 mg DON/kg feed), the presence of DON in chickens feed might have growth inhibitory effects [54,55,61]. A decreased BWG of broilers fed 1.68 mg DON/kg feed at 21 days has been observed [61]. Slow growing chicks fed 2 mg DON/kg feed had lower BWG [54]. The body weight (BW) and feed intake of broiler chickens have shown to be reduced after feeding 2.5 mg DON/kg feed for 35 days [55]. Furthermore, DON exposure from 5 to 14 mg/kg of feed for 35 days had adverse effects on BW, BWG, feed intake, and feed conversion ratio of chickens [27,55,56,62]. Broiler chickens exposed to DON from 15 to 18 mg/kg feed for 21 days also showed lower performance compared with controls, according to several studies [63,64,65,69]. Recently, Riahi et al. [19] also reported that the use of DON artificially contaminated feed (15 mg/kg) during 42 days reduced the BWG and altered the feed conversion ratio. The poor performance occurring in the presence of DON in feed could be related to the mode of action of this toxin, that is principally the inhibition of protein synthesis at the elongation or termination steps [14].

It was shown also that DON adverse effects on chicken’s productive parameters may appear only during specific parts of their growth cycle [51].

Awad et al. [28] noted that the adverse effect of DON was only pronounced at the beginning of the experiment, but no effect was observed later on BW, BWG, feed consumption, and feed conversion ratio. In fact, they found that feeding DON at 1 or 5 mg/kg to broiler chickens decreased the feed intake in a quadratic manner during the first week and BW and BWG during the second week [33]. Moreover, a reduction of feed intake was observed only at the third week of the trial after DON (5 mg/kg) exposure to broilers [49]. Adverse DON temporary effect on growth performance was also observed in chickens fed 1.68 or 12.20 mg DON/kg feed during the 21 days of exposure, and this effect disappear later [61]. BWG was only reduced during the second week in the study conducted by Kubena et al. [66], which used 16 mg DON/kg of feed during 21 days. The meta-analysis conducted by Andretta et al. [72] reported that the effect of mycotoxins on broiler’s growth was greater in young broilers.

It has been hypothesized that the temporal effect produced especially at the early stages of chicken development is related to the capacity of birds to later adapt to DON mycotoxicosis.

In contrast, few studies found that broilers are more sensitive to DON-contaminated diets during later stage of growth. Feeding DON from 5.9 to 9.5 mg/kg feed to broilers reduced BWG and feed intake during the grower period (21 to 42 days) [53]. Broiler chickens fed 7.90 mg DON/kg feed from 21 to 34 days had lower BW, BWG, feed intake, and altered feed conversion ratio [51]. The authors concluded from this study that after chronic exposure to DON-contaminated diets, the adverse effects on growth performance are not necessarily cumulative [51].

However, other reports indicated that DON at low, moderate, high or extremely high levels had no adverse effects on poultry performance. The exposure to naturally contaminated DON diets, containing 1.5–1.87 mg DON/kg to broiler chickens for 21 or 28 days had no effects on BW, BWG, feed consumption, and feed conversion [45,48]. The absence of effect could be due to a level of DON contamination sharply lower than ones adopted in the previous research [51,53]. No adverse effect was indicated on broiler performance feeding the guidance value for 15, 21, 28, or 42 d, respectively [19,46,48,50]. BW, BWG, feed intake, and feed conversion ratio were not adversely affected by the inclusion of 10 mg DON/kg in broiler diets for 35 or 42 days [57,58,59,60]. Productive parameters of birds exposed to 16 mg DON/kg feed for 21 days were not adversely affected [67]. Even high level of an artificially DON contaminated diet (116 mg/kg) did not adversely affect BWG and feed consumption of broiler chickens from 6 to 11 days [71]. Failure to observe significant differences on growth parameters might be suggestive of adaptation of birds to mycotoxins over time, and that poultry are relatively tolerant to DON mycotoxicosis compared with other species, especially pigs, due to the differences in DON absorption, distribution, metabolism, and elimination (ADME) [22].

Furthermore, it has been reported that DON could have growth stimulatory effects. Some studies indicate that DON in poultry feed enhance growth. Feeding 4.6 mg DON/kg feed to Leghorn and broiler chicks for 4 weeks (between 7 and 35 days of age) increased their daily BWG [47]. In the same manner, BW of male Leghorn chicks was increased when fed 9 mg DON/kg feed at 35 days and 18 mg DON/kg feed at 21, 28 and 56 days [68,70].

In addition, the inclusion levels of 4.7 or 8.2 mg DON/kg in broiler chickens feed increased BWG and feed intake in a significant quadratic manner in the finisher period [52]. A contaminated diet with DON at 1.5 mg/kg for 35 days enhanced broiler performance by increasing the BWG and reducing the feed conversion ratio [31]. The enhancement of performance might be the consequence of the increase of nutriments digestibility induced by the presence of the fungus in the grains [62]. Taiwan country chickens exposed to 5 mg/kg of DON in feed during 16 weeks showed a growth promoting impact in comparison with chickens receiving 2 mg DON/kg [54]. It has been suggested that this concentration near to or equal to 5 mg/kg could have a stimulator and promoting effect on growth of broilers and it was described as an hormesis phenomenon [52], resulting in either a J-shaped or an inverted U-shaped dose response.

In summary, the adverse effect of DON on broiler performance is the most frequent effect reported among studies. The impairment of BWG, feed intake, and feed conversion ratio could reach up 37, 25 and 35%, respectively [27].

The improvement of genetics of broiler chickens over the years could be a factor of consideration of growth effects. In fact, rapidly growing birds may have altered metabolic and nutrient partitioning and has long-been criticized in reducing immunity [73]. Then, rapidly growing birds are more susceptible to detrimental effects of mycotoxins on growth. The effects of DON on productive parameters may depend on the dose and the duration of exposure. The source of mycotoxin used (naturally vs. artificially) is considered also a factor of variation of DON toxicity, in which natural contaminated diets with DON are more frequently toxic than artificially contaminated diets, as reported in Table 2. Moreover, in natural contamination, multi-mycotoxin contamination is very frequent, as most of fungi are able to produce several mycotoxins simultaneously, and feed can be contaminated by several fungi species at the same time. Thus, mycotoxin multicontamination of raw materials is more likely to occur than a single mycotoxin contamination [10,74]. Additionally, the presence of acetylated and glycosylated forms of DON in naturally contaminated feed can increase the global toxicity as these forms are cleaved into DON in digestive tract of animals [75]. In addition, the effects of DON on productive parameters may depend on the climatic factors, the methods of analysis of mycotoxins and the adaptability and the tolerance of chickens to DON mycotoxicosis [5,48]. This tolerance may be due to the biotransformation of DON to DON-3S in chickens, which is much less toxic than DON itself [35].

### 3.2. Relative Weight of Organs

The effect of DON on organ weights of chickens is summarized in Table 3.

Regarding DON toxicity, this biomarker is highly variable and results are contradictory. The weight of organs expressed as a percentage of the body weight of the bird is named relative weight (RW). Some organs seemed to be the most affected in different studies as, for example, organs with high turnover of protein such as liver, immune organs and small intestine [14]. The RW of liver decreased in birds fed 18 mg DON/kg feed during 5 weeks [68], but this parameter remained unaltered using the same dose of toxin during 12 weeks, and this may be explained by the difference in the duration of exposure [70]. Indeed, the time of toxin exposure may be a significant factor as the organ initially swells with toxin exposure followed by shrinkage [53]. Similarly, in broiler chickens exposed to DON at different levels (up to 10 mg/kg), a significant decrease of liver RW is observed at week 3 of the trial [55]. However, in another trial, this parameter increased temporally in the second week of exposure for broilers fed a low DON diet (1.68 mg DON/kg) [61]. The changes observed on the RW of the liver might be associated to lipid metabolism alterations [63].

It has been previously observed that 10 mg of DON/kg diet for 35 days decreased the RW of kidneys of broiler chickens and it has been suggested that DON alter the indicated organ cells [76]. In contrast, other reports indicated that the RW of kidneys remained unaltered [62,65,68,70].

The RW of gizzard increased in several studies after DON exposure of broiler chickens [19,64,65,68,70,76]. The increase of RW of gizzard might be directly related to a difference in the density of the diets or might be a result of the irritation of the upper gastrointestinal tract [68].

The RW of small intestine of broilers fed 5 mg/kg DON for 21 days decreased [49]. In the same manner, Yunus et al. [77] reported a reduction in the RW of duodenum and jejunum, and they suggested that this decrease could be due to the reduction of villus height which then resulted in decreased digestion of nutrients and energy.

Regarding the immune organs, different results have been reported using the same DON concentration during the same exposure time; the RW of bursa of Fabricius of birds did not change after feeding 16 mg DON/kg of feed for 21 days [65] and, using the same trial conditions, an increase of the RW of bursa of Fabricius was found [66]. Results of these studies are highly variable, suggesting that organ weights might not be a good indicator of DON toxicity [78]. An increase of the RW of spleen for broilers fed a low level of DON (1.68 mg/kg) at week 4 was reported [61]. Similarly, the RW of spleen was higher in the chickens fed 5 mg/kg of DON contaminated diet and this increase was interpreted as a consequence of the irritation of chicken’s immune system by the toxin, the swelling of this organ and some alterations on a cellular level [54]. Nevertheless, the indicated immune organ was reduced in Lohman chickens fed up 14 mg DON /kg for 5 weeks [62]. In other studies, the RW of spleen was not affected by DON chickens’ diets [19,33,49,53,60,65,68,70,76]. In our recent study, a significant increase of thymus of birds exposed to DON at 5 or 15 mg/kg for 42 days was reported [19].

Due to the contradiction between results, it can be considered that RW of organs might not be a very precise biomarker of DON toxicity in birds. However, based on the frequencies of effects of DON on RW of organs, the RW of gizzard being the one the most impacted by DON, it can be considered as a promising biomarker.

### 3.3. Intestinal Morphology

As all trichothecenes, DON is a small molecule that has the capacity to passively cross through cell membranes and to be easily absorbed by the epithelium cells of the intestine. These epithelium cells can be exposed to high concentrations of DON following ingestion of contaminated diets [21]. On the other hand, the intestinal tract is a very important barrier to ingested feed contaminants and is also the first line of defense against various gut antigens [79]. Moreover, it has been reported that DON alter intestinal morphology [33]. The studies reporting the impairment of intestinal morphology parameters in chickens are listed in Table 4.

The density of the intestine was identified as the ratio between the absolute weight of small intestine and its length. Birds fed DON-contaminated diets from 1 to 21 days had lower intestinal density [51]. In broilers exposed to 1.68 or 12.20 mg/kg for 35 days, DON exposure has shown to produce a reduction of the weights of duodenum and jejunum and an increase of length of both segments and, consequently, DON decreased the small intestine density, in a dose- and time-dependent manner [77]. Additionally, the length was lower, and the density of the small intestine was higher in broilers receiving 5 and 15 mg DON/kg compared with controls [19]. Contaminated DON feed at concentration 3–4 mg/kg, which is below the European maximum guidance level, resulted in a decreased duodenal villus height [80]. Jejunal and ileum histological analysis revealed that broiler chickens exposed to the DON diets for 34 days had shorter villi and reduced crypt depth than control birds [51]. Additionally, it has been shown that chickens consuming DON (5 or 10 mg/kg feed) had shorter and thinner villi compared to chickens ingesting control diets [49,60,82,83]. It has been also reported a significant reduction of villus height, villus surface area and muscularis thickness in jejunum of broiler chickens upon feeding naturally contaminated diets with DON (1 or 5 mg/kg feed) [33]. Data reported by Yunus et al. [77] indicated that feeding 12.20 mg/kg of DON contaminated diet to broiler chickens resulted in a reduced villus height in the duodenum and jejunum. Similarly, feeding DON contaminated feed (7.54 mg/kg) for 21 days decreased the villus height and crypt depth both in duodenum and jejunum of broiler chickens [81]. In the same manner, villus height and villus height to crypt depth ratio in the duodenum were decreased in chickens receiving the guidance value for 4 weeks [50]. Only one study shows opposite results regarding the intestine morphology, as they observed that feeding diets at a concentration up to 18 mg of DON/kg of feed linearly increased broiler villi heights [69]. In recent reports villus height was unaffected by the dietary inclusion of DON [19,54].

Decrease of villus height may be due to villus contraction and leads to the impairment of nutrient transport and utilization, resulting in alteration of gain weight of animals fed DON [84]. Furthermore, Robert et al. [85] indicated that the shortening of villus height produced by DON may be explained by the implication of this mycotoxin in the balance between epithelial cell proliferation and apoptosis. As such, the adverse effect of DON on the villus height can be linked with an impaired nutrient digestion due to a reduced number of differentiated epithelial cells [86].

After a mechanical or toxic damage, more enterocytes need to be generated to migrate progressively along the villus toward the tip which induced the change in crypt depth [87]. In addition, change in crypt depth in birds fed DON could be attributed to the capacity of DON to reduce cell proliferation [81]. The crypt-depth to villus-height ratio was calculated to assess the intestinal architectural changes after the treatment [87]. 

Changes observed in morphology and histology of small intestine could be due to the irritant effects of DON on the upper gastrointestinal tract and the inhibition of protein biosynthesis, leading to the alteration of the absorption and digestion mechanisms which negatively affect bird growth [79,81].

To conclude, we could consider that the decrease of villus height is the most frequent effect reported among studies. This biomarker could be useful in the research on developing and efficacy proving of mycotoxin-detoxifying agents. Moreover, research is now growing in the field of gut health and focusing on the effects of mycotoxins in the gastro-intestinal tract (GIT). 

### 3.4. Biochemical and Hematological Parameters

Biochemical and hematological parameters are used to infer the health status of animals. The most significant results of the effect of DON on biochemical blood parameters are indicated in Table 5.

Dietary DON may affect the enzymes reflecting liver and kidney activities such as lactate dehydrogenase (LDH), alanine aminotransferase or transaminase (ALT), aspartate transaminase (AST), alkaline phosphatase (ALP), gamma-glutamyl transferase (γ-GT), creatine kinase (CK), and creatinine. Broilers exposed to 15 mg DON/kg feed during 21 days had higher AST activity in blood [63]. However, the same dietary level in chickens feeding during 42 days did not affect this enzymatic activity [19]. ALP and AST activities increased in broilers exposed to 2.95 mg DON/kg feed for the second last week of the trial [88]. Faixová et al. [89,90] showed an elevation of serum ALT activity in broilers exposed to 3 mg DON/kg feed for 6 weeks. However, broilers fed DON (10 mg/kg) for 35 days had lower ALT [58]. Swamy et al. [52] reported that DON at 4.6 mg/kg increased the levels of γ-GT activity. These results were in accordance with a previous report of chickens fed 15 mg DON/kg [63]. Contaminated diet with DON at 5 mg/kg decreased the serum creatine kinase level in broiler chickens aged 42 days [19], reflecting the reduction of this enzyme from the circulation [93]. Generally, the changes obtained in enzymatic activities might be due to hepatic disorders, chronic liver damage, leakage of the enzymes into the blood, biliary obstruction, or kidney affliction [52,58,88].

For biochemical parameters reflecting lipid metabolism, higher cholesterol level and triglycerides were observed in broilers fed DON at 10 mg/kg for 35 days [58]. The increased cholesterol level may suggest liver or kidney damage function and a high stress status of the bird [58]. The increased level of triglycerides might be related to lipid metabolism alteration and or to biliary obstruction [58]. These results are in contrast with the previous and actual findings which observed decreased cholesterol and triglycerides levels in broilers [19,27,64,68,89,90]. This is in agreement with the meta-analysis report of Andretta et al. [94] indicating that broilers fed challenged mycotoxins (T2, FBs, DON, OTA, and ZEN) had lower total cholesterol (−14 %) and lower triglycerides (−39 %) compared to negative control birds. It was suggested that these changes in indicated biochemical parameters could be explained by involvement of liver and a shift of concentrations from the blood to the liver [95].

Regarding biochemical parameters reflecting protein metabolism, it has been described that the total plasma protein and albumin levels of chickens fed a contaminated diet with 2.95 mg DON/kg feed during 2 weeks of exposure were decreased compared to control [88]. The reduction of total protein level after feeding DON to broilers was observed also in previous reports [27,89,90]. This decrease could be the result of protein and DNA synthesis inhibition induced by DON [14]. Additionally, the decreased plasma protein level may be related to the decrease of feed intake. DON toxicity is expressed also through a change in the profile of uric acid (UA) [27,52]. The decreased level of UA may be attributed to the efficiency of amino acid utilization, changes in enzyme systems, altered renal filtration, and reabsorption rates [95]. 

Blood hematological parameters serve as indicators of the physiological state of birds [96], but very limited information is available regarding the effect of DON exposure on hematological variables in poultry. The absence of significant effect of DON exposure on hematology of broiler chickens exposed to different DON concentrations (up to 16 mg/kg feed) at 21 or 35 days has been shown [62,67]. Even 50 mg/kg of DON did not change these indicators in broiler chickens aged 21 days [92]. In other studies, a significant decrease in hemoglobin concentration was observed accompanied or not with a decrease on hematocrit and a decrease on erythrocytes count of birds exposed to DON mycotoxin [52,64,65,70,91,92]. In their meta-analysis, Andretta et al. [94] reported that the presence of mycotoxins in broiler diets decreased the hematocrit and the hemoglobin concentration by 5 % and 15 %, respectively. The decreasing of hematocrit hypothesized that hematopoietic tissue may be affected by DON toxin [97]. However, no hemorrhage or hemolysis was observed in broilers. The decrease of hemoglobin and erythrocytes does not necessarily mean that DON induces anemia in poultry, because values are within the range of reference [96]. Moreover, the decrease of hemoglobin and erythrocytes is a marker of immune system alteration. 

The most frequent effect of DON in blood biochemistry and hematology of chickens among studies was the decrease of blood protein, triglycerides, hemoglobin, and erythrocytes and the increase of ALT. These indicators could be useful to evaluate the DON effect on the physiological and hemostatic state of chickens. However, it should be noted that blood biochemical and hematological parameters were slightly affected by DON contaminated feed, as these parameters were still within their physiological reference range [98,99,100], and therefore, of limited physiological relevance.

### 3.5. Biomarkers Related to Immune System

Few studies have been conducted regarding DON effects on poultry immune system, despite that the first outcome of DON toxicity is immune system functions and responses. As the immune system has rapidly proliferating cells and tissues with high rates of protein turnover, it is consequently considered the most susceptible system to DON and trichothecenes mycotoxicosis [14]. Responses to common vaccines as Newcastle disease virus (NDV) and infectious bronquitis virus (IBV) are considered as humoral immune response parameters [27,62]. Leukocytes apoptosis and lymphocytes proliferation are considered as cellular immune responses to DON detrimental effects [27,101]. Furthermore, the production of proinflammatory cytokines could be used as an indicator to evaluate the effects of DON contaminated diets exposure on immune system competence or modulation [102]. The effect of DON on immune system functions and responses through the evaluation of those indicators mentioned above are reviewed in Table 6. 

#### 3.5.1. Biomarkers Related to Humoral Immune Response 

Dänicke et al. [62] suggested that the antibody titers against NDV could be a relevant biomarker for DON effects on protein synthesis inhibition due to the reduction of antibody titers to NDV when broiler chickens fed up 14 mg of DON/kg feed for 5 weeks. In the same manner, broilers exposed to a mixture of *Fusarium* mycotoxins, included DON, presented a decreased NDV titers at 28 and 42 days of age [104]. Additionally, NDV antibody titers decreased in White Leghorn chicks fed 18 mg of DON/kg of feed at 18 weeks [92]. However, the exposure to 12.2 mg of DON/kg of feed of broilers for 5 weeks increased titers against NDV at week 2 and week 4 but decreased numerically at week 5 [61]. As trichothecenes, DON is both immunostimulatory and immunosuppressive, depending on the level and the length of exposure [101]. The enhancement or the depression of the response to NDV observed may be due to the ability of DON to modulate the immune response during chronic exposure. Concerning the antibody titers against IBV, it was concluded that it is also a good indicator of depression of protein synthesis induced by the exposure of DON [27]. Antibody titers to IBV decreased in broiler chickens receiving DON at 10, 12.2, or 12.6 mg/kg for 5 weeks [27,61]. Meanwhile, no effect on the IBV titers was found following exposure of broiler chickens to different concentrations of DON (4.7, 8.2, or 9.7 mg/kg of feed) for 21 and 42 days [52]. Furthermore, no effects of a concentration of 12.2 mg DON/kg were found on antibody against IBV titers in broiler chickens after 14 and 28 days [61]. 

The discrepancies in results of these biomarkers of effect might indicate that the humoral immune response of broilers fed DON is variable, and further studies should be undertaken [61]. It might be interesting for these studies to take into account parent breeder vaccination schedule and chick maternal antibody levels. 

#### 3.5.2. Biomarkers Related to Cellular Immune Response 

The first molecular target of DON and other trichothecenes is the ribosome. DON inhibits protein synthesis via binding to the peptidyl transferase region of the 60S subunit of the ribosome and interfering with the elongation step [107]. Ribosome binding leads first to the activation of ribosomal-associated protein kinase R (PKR) and tyrosine protein kinase (Hck) and, subsequently, the activation of mitogen activated protein kinases (MAPKs) signaling, including p38, Jun N-terminal kinase (JNK) and extracellular signal-regulated kinase 1 and 2 (ERK1/2) [108]. The mechanism regulating this activation is named the “ribotoxic stress response” [107]. The activation of MAPKs pathway by DON in vivo and in vitro suggests that the ribotoxic stress response mediates DON toxicity [109]. MAPKs are involved in cell growth, differentiation and apoptosis [108]. Additionally, it might include leukocyte apoptosis and lymphotoxicity [110].

Leukocytes are target to DON and trichothecenes mycotoxicosis [110]. As described previously, DON is both immunostimulatory and immunosuppressive and it has been reported that at high doses, DON is immunosuppressive and causes leukocyte apoptosis [111]. The apoptosis was induced by the activation of c-Jun terminal kinase, p38 mitogen-activated protein kinase and caspases. At high doses, trichothecenes produced injuries in spleen, thymus, bone marrow, and intestinal mucosa, leading to immunosuppression, and potentially increased susceptibility to several pathogens [112]. Exposure to high DON concentrations induces macrophages apoptosis which can lead to innate immune function suppression [101]. Moreover, the acute oral exposure to trichothecene mycotoxins produced adverse effects to actively dividing cells in bone marrow, lymph nodes, spleen, thymus, and intestinal mucosa [110].

Trichothecenes have both effects of enhancement or impairment of mitogen-induced lymphocyte proliferation, depending on the dose of exposure, what has been demonstrated in in vitro and ex-vivo studies [111]. In vitro DON exposure induced apoptosis in murine T cells, B cells, and IgA+ cells isolated from spleen, Peyer’s patches, and thymus [113]. This induction of apoptosis was dependent on lymphocyte subset, tissue source, and glucocorticoid induction [110]. In vitro lymphotoxicity of type B and type A trichothecenes depends on the degree of acylation in substituent groups, according to Pestka et al. [111]. Furthermore, Sharma [114] noted that lymphocytes cytotoxicity is a result of a DON immunotoxic effect. DON has the ability to promote in vitro apoptosis in chicken spleen lymphocytes and tissue [54,115]. Ingestion of DON from 5.9 to 9.5 mg/kg feed reduced the B cells and T cells counts in broiler chickens [53]. In the in vivo study conducted by Ghareeb et al. [27] blood lymphocytes count was decreased in broilers receiving DON (10 mg/kg feed) for 35 days.

The observed changes on leukocytes and lymphocytes in mycotoxicosis conditions can be attributed to the immune system being sensitive to DON [10].

#### 3.5.3. Production of Proinflammatory Cytokines

After MAPKs activation, p38 and ERK induced gene transcription and enhanced mRNA stability [107]. The induction of genes expression included also the induction of transcription and post transcriptional factors, such as nuclear factor kβ (NF-kβ) and leads to promote proinflammatory genes [107,108]. Proinflammatory cytokines might serve as relevant biomarkers of effect that are predictive of DON’s negative effects in target animals.

The gene expression of interferon gamma (IFN-γ) in the cecal tonsils of chickens fed *Fusarium* mycotoxins challenged with coccidian was upregulated [105]. Broiler chickens exposed to a mixture of mycotoxins, in which DON concentration was approximately 2 mg/kg, for 42 days had higher expression of interleukin 1-β (IL1-β) and IL-6 and lower expression of IFN-γ in spleen tissues [104]. In a recent study conducted by Lucke et al. [102], the gene expression of IL-6 was increased in broilers receiving up 5 mg/kg for 35 days. Similarly, DON at 5 mg/kg feed upregulated the relative gene expression of IL-6, IL1-β and IFN-γ in broiler chickens at 42 days.

The upregulation of cytokine gene expression by DON was explained by the ability of this mycotoxin to inhibit synthesis of labile protein repressors of mRNA expression [105,116]. Zhou et al. confirmed that the induction of gene expression of cytokines is a result of DON-induced transient expression of specific mRNAs responsible for impairment of synthesis of high turnover proteins [117]. It has been suggested also that this induction of immune related genes by DON mycotoxin is associated to a higher transcription factor’s binding activity in leukocytes at the transcription level and to a higher mRNA stability at the post-transcription level [101,105]. In addition to that, the upregulation of cytokines production is linked to the transcription factor NF-κB activation [108]. 

However, in response to DON (10 mg/kg feed), broiler chickens had lower LITAF in plasma and lower IL-1β, IFN- γ and TGFBR1 in jejunum, and no effect was observed on plasma IL-8 and LITAF nor on IL-8 and NF-kB1 in the jejunum [106]. Grenier et al. [103] reported a downregulation of IL-6 in cecal tonsils in broiler fed 1.6 mg DON/kg feed. No effect on LITAF and IL-1β in spleen and bursa of Fabricius tissues was observed in broilers exposed to 5 mg/kg DON for 28 days [50]. The differences obtained among studies regarding the DON effect on cytokines production depends on the biological matrix. For example, the determination of cytokines in serum could be a nonsuitable indicator to evaluate the cytokine production due to their short-half-life or due to the low sensitivity of the method used [117]. Another explanation could be the modulating effect of DON on the innate immune response [106]. The ability of DON to upregulate cytokines release is associated to the effect of DON on the innate immune response, leading to the impairment of chicken’s resistance to infectious disease [106]. Furthermore, DON-induced proinflammatory cytokines can be useful in the underlying cause of DON-induced feed refusal. Infection induced anorexia could be explained by the upregulation of relative gene expression of IL-1β andIL-6 [118].

### 3.6. Biomarkers Related to Welfare Parameters

To evaluate welfare related parameters induced by DON toxicity in chickens, the indicators linked with oxidative stress and to physiological, hormonal, and behavioral welfare are highlighted in Table 7.

#### 3.6.1. Response to Oxidative Stress as Welfare Biomarker

DON induces oxidative stress, which is defined as the imbalance between the reactive oxygen species (ROS) and the antioxidant defenses [125]. DON exposure to chicken’s exposure to DON induced the overproduction of free radicals, such as the upregulation of the hypoxia inducible factor 1, subunit alpha (HIF-1α) and heme-oxygenase (HMOX) in jejunum and xanthine oxidoreductase in the liver of broiler chickens exposed to 7.54 mg DON/kg contaminated feed for 21 days [81]. An increased production of ROS leads to DNA damage, protein oxidation and lipid peroxidation [126]. The exposure of broiler chickens exposed to DON (10 mg/kg feed) resulted in DNA damage in spleen leukocytes or in blood lymphocytes [76,121,122]. Regarding the parameters associated with lipid peroxidation, it has been reported that DON increased thiobarbituric acid reactive substances (TBARS) in jejunum and malondialdehyde (MDA) in liver, kidney, serum, and jejunum [76,83,120,123]. On the other hand, the oxidative stress is characterized by a reduction in cellular antioxidant levels with a decreased activity of antioxidant enzymes such as superoxidase dismutase (SOD), catalase (CAT) and glutathione-related systems (GSH), including glutathione peroxidase (GPx) [126]. In vivo and in vitro studies reported lower levels of SOD in serum [83], GSH in the liver of broilers [120], and both parameters in chicken embryo fibroblast DF-1 cells after DON exposure [119]. The ingestion of DON reduced the GSH in liver, jejunum and ileum, the 2,2’-azino-bis (3- ethylbenzothiazoline-6-sulphonic acid) (ABTS) in the jejunum, and the ferric reducing ability (FRAP) in liver of broilers chickens [123]. Oxidative stress data may constitute an effective source of information to know the bird’s welfare assessment.

#### 3.6.2. Biomarkers Related to Physiological, Hormonal, and Behavioral Welfare

Only a few studies have been carried out on the impact of DON contamination of poultry feed on biomarkers related to physiological, hormonal, and behavioral welfare. Onbasilar and Aksoy [127] reported that the impairment of the number of circulating leukocytes, such as heterophils and lymphocytes, is part of the physiological stress. Therefore, the heterophil to lymphocyte ratio (H/L) was used as a good indicator to evaluate the stress in poultry [128]. 

The stress index was elevated in broilers fed DON (10 or 18 mg/kg) for 21 or 35 days [27,57,69]. However, Dänicke et al. [62] did not report significant differences on H/L ratio in broilers fed a contaminated diet with a concentration of 14 mg/kg of DON per kg. Likewise, the H/L ratio was higher in broilers fed 5 or 15 mg of DON/kg of feed, but this elevation was not statistically significant [91]. 

The elevation of circulating levels of corticosterone is included also within the physiological stress biomarkers [127]. Antonissen et al. [46] showed that broilers exposed to a DON (4.6 mg/kg) contaminated diet for 15 days had higher mean plasma corticosterone levels than chickens fed control diet. Similarly, Ghareeb et al. [57] observed that feeding broilers with 10 mg/kg of DON for 35 days, increased the plasma corticosterone level. The elevation of corticosterone was related with the upregulation of gene expression of IL-1β and IL-6 proinflammatory cytokines [129]. The increase in corticosterone might be associated with the mycotoxin induced increase in susceptibility to infectious diseases [80].

The fear response is a welfare-related behavior evaluated by tonic immobility reaction and can provide more information on chicken’s stress status [57,130]. The duration of tonic immobility was longer in broilers exposed to 10 or 15 mg DON/kg [57,91]. This effect could be related to the adverse effect of DON on brain regional neurochemistry. Indeed, the inclusion of contaminated grains with *Fusarium* mycotoxin in the diet of broiler chickens for long-term (1–56 days) increased the concentration of 5-hydroxytryptamine (5 HT) in the pons and in the cortex, which is responsible for the regulation of fear [124]. DON-induced stress altered brain neurochemistry of broilers by increasing serotonin levels (strong satiety neurochemical), suggesting that DON caused partial feed refusal and growth suppression [124]. Consequently, these adverse effects on bird’s welfare might be associated with the mycotoxin induced increase in sensitivity to infectious diseases. 

In summary, the frequencies of effects of biomarkers mentioned in this review are recapitulated in Table 8.

It should be noted that the most frequent effect on these biomarkers appeared at 42 days (Table 8), indicating that sampling time and measurements of biomarkers at 42 days could be suitable to evaluate toxic DON effect on broilers, due to the fact that older birds consume more total mycotoxins, which is estimated by feed intake (kg) x dietary mycotoxin levels (mg/kg) [51]. Moreover, the differences in absorption, distribution, metabolism and excretion processes could be age related and then could influence animal’s response to mycotoxins [126].

## 4. Conclusions

To conclude, DON-3S is the specific and the most suitable biomarker of exposure of DON in chicken. The metabolic pathway that causes its formation could be considered as a detoxification pathway in those animals, which probably explains the low susceptibility to DON mycotoxicosis compared to other species. However further studies are needed to elucidate exactly the metabolism leading from DON to DON-3S in poultry.

Among the biomarkers of effect highlighted in this review, the most relevant ones are productive parameters, relative weight of gizzard, villus height, blood protein, triglycerides, alanine transaminase, hemoglobin, erythrocytes, lymphocytes, response to NDV and IBV, and stress index (H/L ratio). Intestinal cytokines (IL-6, IL-8, IL-1β, and IFN-γ), and TBARS as an oxidative stress parameter could be also relevant biomarkers of toxicity of DON in chickens.

DON could alter the production and bird’s health status through multiple pathways. The biochemical and hematological alteration suggests the malfunctioning of immune system. Furthermore, the intestinal alteration leads to the alteration on the absorption and digestion process. The induction of proinflammatory release, corticosterone secretion and oxidative stress, are involved in feed refusal and increased the susceptibility to infectious disease. Studies cited in the current review were under experimental conditions, however, it would be necessary to increase the proportion of research undertaken in field conditions.

## Figures and Tables

**Table 1 toxins-13-00217-t001:** Biomarkers of exposure of deoxynivalenol (DON) in chickens.

DON ^1^ Dose	Route	Matrix	Metabolites ^2^ Analyzed	Analysis ^3^ Method	LOQ (ng/mL or ng/g) ^4^	Main Metabolite	Reference
**Acute or chronic administration of DON (farm studies)**							
1.5 mg/kg	Feed	Plasma, bile, liver and breast meat	DON DOM-1	HPLC with diode array detection	6.6 (plasma) 13.2 (bile, liver, and breast meat)	-	[31]
9.5 mg/kg	Feed	Plasma	DON DOM-1	LC–ESI-MS/MS	0.1 for DON 0.2 for DOM-1	DON DOM-1	[32]
1 or 5 mg/kg	Feed	Serum, bile, liver, digesta of (gizzard, cecum, rectum), and excreta	DON DOM-1	HPLC-MS/MS	23.3	DON	[33]
2.44 or 7.54 mg/kg	Feed	Plasma, liver, kidney, bile	DON DOM-1	LC–MS/MS	1.25	DON (plasma and bile) DOM-1 (bile)	[34]
2.5 mg/kg BW	Oral administration	Plasma and organs ^5^	DON 10-DON-sulfonate 10-DOM-sulfonate DON-3S	Radiotracer method coupled (γ-ARC) (radio-HPLC-IT-TOF-MS/MS)		DON-3S	[35]
1.7 mg/kg	Feed	Excreta	DON-3S DOM-3S DON, DOM, DON sulfonates 1,2,3, and DOM sulfonate 2	LC–HR-MS/MS	1 for DON sulfonates 1,2,3, and DOM sulfonate 2 4.5 for DON-3S DOM-3S DON and DOM	DON-3S	[36]
0.5 mg/kg BW	Single intra-crop bolus	Plasma and excreta	DON DON-3S	LC–MS/MS and HR-MS	1	DON-3S	[37]
**Toxicokinetics studies (laboratory studies)**							
0.75 mg/kg BW	Intravenous injection or oral gavage	Plasma	DON DOM-1	LC–MS/MS	1–2.5	DON	[38]
0.5 mg/kg BW	Intravenous injection or oral gavage	Plasma	DON 3 ADON 15 ADON DOM-1	LC–MS/MS	0.1-2	DON	[22,39]
0.75 mg/kg BW	Intravenous injection or oral gavage	Plasma	DON DOM-1 DON-3S DON-3G 10-DON-sulfonate, DOM-1 and 10-DOM-1-sulfonate	LC–MS/MS	0.1	DON-3S	[30]
0.77 DON-3G mg/kg BW 0.5 DON mg/kg BW	Intravenous injection or oral gavage	Plasma	DON DON-3G DOM-1	LC–MS/MS and HR-MS	1	DON-3S	[40]

^1^ DON, deoxynivalenol; ^2^ DOM-1, deepoxy-deoxynivalenol; ADON, acetyl-deoxynivalenol; DON-3S, deoxynivalenol-3-sulphate; DON-3G, deoxynivalenol-3-glucuronide; ^3^ HPLC, high performance liquid chromatography; LC–ESI-MS/MS, liquid chromatography–electro-spray ionization tandem mass spectrometry; LC–MS/MS, liquid chromatography–mass spectrometry; radio-HPLC-IT-TOF-MS/MS; radiotracer method coupled with a novel γ-accurate radioisotope counting (γ-ARC) radio-high-performance liquid chromatography ion trap time-of-flight tandem mass spectrometry system; ^4^ ng/mL (fluids) or ng/g (organs and excreta); ^5^ gastrointestinal tract, plasma, bile, brain, heart, liver, kidney, lung, spleen, pancreas, urinary bladder, ovary/testis, adipose, muscle, and skin.

**Table 2 toxins-13-00217-t002:** Effects of deoxynivalenol (DON) on productive parameters of chickens.

Breed	DON ^1^ (mg/kg) Diet	Source of Contamination	Exposure Duration (day)	Reported Effects ^2^	Percentage of Variation %	Reference
Lohmann Meat	1.5	Artificial	35	↑ BWG ↓ feed conversion ratio	8 10	[31]
Shaver	up to 1.87	Artificial	28	None	-	[45]
Ross 308	4.6	Artificial	15	None	-	[46]
Leghorn chicks (Ottawa strain 10)	4.6	Natural	28 (from day 7 to 35)	↑ BWG ↑ feed intake	2 4	[47]
Ross 308	1 or 5	Natural	35	↓ feed intake (7 days) (1 mg/kg) ↓ BW (14 days) ↓ BWG (14 days)	19 26 or 5 33 or 3	[33]
Cobb × Cobb 500	1.5 or 5	Artificial	21	None	-	[48]
Ross E032	5	Natural	21	↓ feed intake (14–21 days)	20	[49]
Ross 308	5	Artificial	28	None	-	[50]
Ross 308	7.90	Natural	34	↓ BW ↓ BWG ↓ feed intake ↑ feed conversion ratio (21–34)	10 14 4 2	[51]
Ross × Ross, Maple Leaf Poultry	4.7 or 8.2	Natural	56	↑ feed intake ↑ BWG At 4.7 (42 to 56 days)	4 2	[52]
Ross × Ross, Maple Leaf Poultry	5.9 or 9.5	Natural	56	↓ BWG ↓ feed consumption (day 21 to 42)	9 or 12 8 or 14	[53]
Black-feathered Taiwan country chickens	2, 5 or 10	Artificial	112	BWG at 2 mg/kg lower than BWG at 5 mg/kg	8	[54]
Ross 308	2.5, 5 or 10	Artificial	35	Overall the trial: ↓ BW at 2.5 and 5 mg/kg, ↓ BWG at 5 mg/kg ↓ feed intake at all doses. At week 5: ↓ BW at 5 mg/kg and ↓ feed intake at 5 and 10 mg/kg	5 or 6 7 7–8 8 12	[55]
Ross 308	5 or 10	-	35	↓ BW ↓ BWG (28 days) ↓ feed intake (28 days) ↑ feed conversion ratio (14 days)	11 12 10 14	[56]
Ross 308	10	Artificial	35	↓ feed intake ↓ BW ↓ BWG ↑ feed conversion ratio	25 17 37 35	[27]
Ross 308	10	Artificial	35	None	-	[57]
Ross 308	10	Artificial	35	None	-	[58]
Ross E032	10	Artificial	42	None	-	[59]
Ross E032	10	Artificial	42	None	-	[60]
Ross 308	1.68 or 12.20	Artificial	35	↓ feed intake ↓ BWG (21 days)	8 or 13 12 or 17	[61]
Lohmann broilers	Up to 14	Natural	35	↓ feed intake ↓BW ↑ in feed to gain ratio	8 4 6	[62]
Ross 308	5 or 15	Artificial	42	↓ BWG ↑ in feed conversion ratio at 15 mg/kg	6 7	[19]
-	15	Natural	21	↓ BWG ↑ in feed conversion ratio	19 28	[63]
Hubbard × Hubbard	16	Natural	21	↓BWG ↑ feed conversion	10 23	[64]
Hubbard × Hubbard	16	Natural	21	↓BWG ↑ feed conversion	9 20	[65]
Hubbard × Hubbard	16	Natural	21	↓BW ↑ feed conversion (day 8 to 14)	2 24	[66]
-	16	Natural	21	None	-	[67]
Leghorn chicks	9 or 18	Natural	35	↑ BW (at 18 mg/kg day 21) ↑ BW (at 9 mg/kg day 35)	7 8	[68]
Ross 708	Up to 18	Natural	21	↓ feed intake ↓ BWG	8 10	[69]
Leghorn chicks	18	Natural	84	↑ BW at 28 and 56 days	5 and 8	[70]
White Mountain 6" × Hubbard 9	Up to 216	Natural	From 6 to 11	None	-	[71]

^1^ DON, deoxynivalenol; ^2^ BW, body weight; BWG, body weight gain; ↑, increase; ↓, decrease.

**Table 3 toxins-13-00217-t003:** Effects of deoxynivalenol (DON) on relative organ of weights of chickens.

DON ^1^ (mg/kg) Diet	Exposure Duration (d)	Reported Effects ^2^	Reference
Up to 1.87	28	No effect on crop, proventriculus, gizzard, intestines, heart, liver, pancreas, kidneys, testes, adrenals, and thyroids	[45]
1 or 5	35	No effect on RW of heart, proventriculus, gizzard, pancreas, liver, small intestine, cecum, colon, thymus, spleen and bursa of Fabricius	[33]
5	21	↓ RW of small intestine, = RW of heart, gizzard, pancreas, caecum, colon, and spleen	[49]
5.9 or 9.5	56	No effect on RW of liver, kidney, spleen, and bursa of Fabricius	[53]
2, 5 or 10	112	↑ RW of spleen (at 5 mg/kg)	[54]
2.5, 5 or 10	35	↓ RW of liver	[55]
10	35	No effect on RW of bursa of Fabricius, spleen, and thymus	[27]
10	35	↑ RW of gizzard, ↓ RW of kidneys, = RW of brain, heart, pancreas, liver, lung, thymus, spleen, and bursa of Fabricius	[76]
1.68 or 12.20	35	↑ RW of liver and spleen, = RW of heart and thymus	[61]
1.68 or 12.20	35	↓ RW of duodenum and jejunum, = RW of proventriculus and gizzard	[77]
Up to 14	35	↑ RW of heart, ↓ RW of spleen, = RW of proventriculus, gizzard, liver, kidneys, small intestine, and bursa of Fabricius	[62]
5 or 15	42	↑ RW of gizzard and thymus, ↓ RW of colon and small intestine, = RW of heart, proventriculus, pancreas, liver, kidneys, cecum, spleen and bursa of Fabricius	[19]
15	21	↑ RW of gizzard, heart and bursa of Fabricius, = RW of proventriuclus, liver and kidney	[63]
16	21	↑ RW of gizzard, = RW of proventriculus, spleen, liver, and kidney	[64]
16	21	↑ RW of gizzard, = RW of proventriuclus, liver, kidneys, spleen, and bursa of Fabricius	[65]
16	21	↑ RW of gizzard and bursa of Fabricius, = RW of proventriculus, pancreas, liver, kidneys, and spleen	[66]
16	21	No effect on RW of liver, left kidney, heart, spleen, pancreas, proventriculus, gizzard, and bursa of Fabricius	[67]
9 or 18	35	↓ RW of liver, = RW kidney, heart, proventriculus, testes, spleen, bursa of Fabricius, and ↑ RW of gizzard	[68]
18	84	↑ RW of gizzard, = RW liver, kidney, heart, proventriculus, spleen, bursa of Fabricius, and testes	[70]

^1^ DON, deoxynivalenol; ^2^ RW, relative weight; ↑, increase; ↓, decrease; =, unaffected.

**Table 4 toxins-13-00217-t004:** Effects of deoxynivalenol (DON) on morphological parameters of small intestine of chickens.

DON ^1^ (mg/kg) Diet	Exposure Duration (Days)	Reported Effects	Reference
1 or 5	35	↓ villus height of jejunum ↓ villus surface area	[33]
2.88 to 4.38	23	↓ villus height of duodenum	[80]
5	21	↓ height and width of villi in duodenum	[49]
5	28	↓ villus height and villus height to crypt depth ratio in the duodenum	[50]
7.54	21	↓ villus height and crypt depth in the duodenum and jejunum	[81]
6.62 to 7.90	21	↓ density of small intestine, ↓ villus height in the jejunum, ↓ villus height, and crypt depth in the ileum	[51]
2, 5, or 10	112	= villus height of jejunum and ileum	[54]
10	21	↓ villus height, ↑ crypt depth, ↓ villus height to crypt depth ratio in the duodenum	[82]
10	35	= length and density of different segments of gastrointestinal tract	[58]
10	42	↓ villus height of jejunum, = crypt depth and villus height and crypt depth	[83]
10	42	↓ villus height in the duodenum and jejunum	[60]
1.68 or 12.20	35	↓ villus height, ↑ length of duodenum and jejunum	[77]
5 or 15	42	↑ length and ↓ density of small intestine, = villus height and crypt depth	[19]
Up to 18	21	↑ villus height in the mid-ileum No effects on crypt depth and goblet cells per villi counts	[69]

^1^ DON, deoxynivalenol; ↑, increase; ↓, decrease; =, unaffected.

**Table 5 toxins-13-00217-t005:** Effects of deoxynivalenol (DON) on blood biochemistry and hematology of chickens.

DON ^1^ (mg/kg) Diet	Exposure Duration (Days)	Reported Effects ^2^	Reference
2.95	28	↓ levels of Tot Prot, Alb ↑ALT, AST and ALP.	[88]
3	42	↓ levels of Tot Prot, mg, Trig and free glycerol, ↑ ALT activity	[89]
3	42	↓ Tot Prot, Trig and free glycerol, ↑ ALT activity	[90]
4.7 or 8.2	56	Quadratic responses in serum concentrations of Alb and γ-GT, ↓ lipase activity, ↑ UA, hemoglobinemia and erythrocytosis	[52]
10	35	↓ of plasma Tot Prot and UA, ↓ plasma Trig level (tendency),	[27]
10	35	↓ level of ALT, ↑ serum Chol and Trig	[58]
Up to 14	35	= Glu, Tot Prot, Hct and Hgb	[62]
5 or 15	42	↓ CK (at 5 mg/kg) and ↓ level of Chol (at 15 mg/kg)	[19]
5 or 15	42	↓ Hgb (dose-dependent), ↓ erythrocytes (at 15 mg/kg)	[91]
15	21	↑ activities of AST, LDH and γ-GT.	[63]
16	21	↓ Trig, alterations in blood erythrocyte count, Hgb and Hct	[64]
16	21	↓ Glu level	[65]
16	21	No effect on Glu, ALT, AST, Creat and Hgb	[67]
9 or 18	35	↓ Glu, Trig and LDH, ↑ Creat, ↓ in Hgb and Hct	[68]
18	63	↓ Hgb concentration, erythrocyte count and Hct	[92]
18	84	↓ in Hgb concentration at 28 days	[70]
50	21	No effect on hematological parameters	[92]

*^1^* DON, deoxynivalenol; ^2^ Tot Prot, total protein; Alb, albumin; UA, uric acid; Chol, cholesterol; Trig, triglycerides; ALT, alanine transaminase; AST, aspartate transaminase; ALP, alkaline phosphatase; γ-GT, gamma-glutamyl transferase; LDH, lactate dehydrogenase; Glu, glucose; Creat, creatinine; CK, creatine kinase; Hgb, hemoglobin; Hct, hematocrit; MCHC, mean corpuscular hemoglobin concentration; ↑, increase; ↓, decrease; =, unaffected.

**Table 6 toxins-13-00217-t006:** Effects of deoxynivalenol (DON) on immune biomarkers of chickens.

DON ^1^ (mg/kg) Diet	Exposure Duration (Days)	Reported Effects ^2^	Reference
1.6	34	↓ IL-6, = IFN- γ, IL-1β, IL-17, and IL-10 in cecal tonsils	[103]
2	42	↓ NDV titers at 28 days and 42 days, ↑ mRNA expression of IL-lß and IL-6, ↓ mRNA expression of IFN-γ in spleen	[104]
Up to 3.8	70	↑ IFN- γ gene expression in cecal tonsils	[105]
Up to 5	35	↑ mRNA expression of IL-6 in the duodenum, ↓ mRNA expression of IL-8 and IL-10 in the jejunum (quadratic trend)	[102]
5	28	No effect on LITAF and IL-1β in spleen and bursa of Fabricius tissues	[50]
4.7 or 8.2	56	= IBV	[52]
5.9 or 9.5	56	↓ B cells and T cells	[53]
2, 5 or 10	112	Apoptosis in chicken spleen lymphocytes	[54]
10	35	↓ of lymphocytes and ↓ IBV	[27]
10	35	In plasma: ↓ LITAF, = IL-8, in jejunum: ↓ IL-1β, IFN-γ, TGFBR1, = LITAF, IL-8 and NF-kB1	[106]
10	35	↓ IBV titers	[58]
12.2	35	↑ NDV (14 days/28 days), ↓ IBV (35 days)	[61]
3.5 to 14	35	↓ NDV titers	[62]
5 or 15	42	= NDV and IBV, ↑ plasma IL-8, ↑ the mRNA of IL-6, IFN- γ and IL-1β in jejunum tissues (at 5 mg/kg)	[91]
18	63	No effect on NDV	[92]
18	126	↓ NDV	[92]

*^1^* DON, deoxynivalenol; ^2^ NDV, Newcastle disease virus; IBV, infectious bronquitis virus; IFN-γ, interferon gamma; IL-1β, interleukin 1-β; IL-6, interleukin 6; LITAF, lipopolysaccharide-induced TNF-α factor, TGFBR1, transforming growth factor beta receptor I; NF- kβ, nuclear factor kappa β; CCK-8, cell counting kit-8; IL-10, interleukin 10, IL-17, interleukin 17; ↑, increase; ↓, decrease; =, unaffected.

**Table 7 toxins-13-00217-t007:** Effects of deoxynivalenol (DON) on welfare related parameters in chickens.

DON ^1^ (mg/kg) Diet	Exposure Duration (Days)	Reported Effects ^2^	Reference
**Response to oxidative stress as welfare biomarker**			
100–2000 ng/mL	24 h	↑ ROS and MDA, ↓ GSH and SOD in embryo fibroblast DF-1 cells	[119]
7.54	21	↑ HIF-1α and HMOX in jejunum ↑ xanthine oxidoreductase in liver	[81]
3.4 or 8.2	14	↑ MDA in liver, kidney and serum, ↓ GPx activity in liver tissue	[120]
10	35	↑ TBARS in jejunum DNA damage in blood lymphocytes	[76]
10	42	↓ SOD activity in serum and ↑ MDA or TBARS in the jejunal mucosa	[83]
10	17	↑ DNA fragmentation in spleen leukocytes No effect on plasma and liver MDA	[121]
10	35	↑ blood lymphocyte DNA damage No effect on TBARS	[122]
19.3	14	↑ TBARS in jejunum No effects on superoxide anion levels of jejunum and ileum ↓ GSH and ABTS in jejunum	[123]
**Biomarkers related to physiological, hormonal, and behavioral welfare**			
Up to 14	35	No significant effect on H/L ratio	[62]
4.6	15	↑ plasma corticosterone	[46]
5.9 or 9.5	56	Alterations on brain neurochemistry ↑5-hydroxytryptamineand serotonin	[124]
10	35	↑ H/L ratio	[27]
10	35	↑ plasma corticosterone, H/L ratio, and duration of tonic immobility reaction.	[57]
5 or 15	42	↓ plasma corticosterone No effect on H/L ratio ↑ duration of tonic immobility reaction	[91]
Up to 18	21	↑ H/L ratio	[69]

^1^ DON, deoxynivalenol; ^2^ MDA, malondialdehyde; GPx, glutathione peroxidase; TBARS, thiobarbituric acid reactive substances, HIF-1α, hypoxia inducible factor 1, subunit alpha; HMOX, heme-oxygenase; ROS, reactive oxygen species; GSH, glutathione-related systems; SOD, superoxidase dismutase; ABTS, 2, 2’-azino-bis (3- ethylbenzothiazoline-6-sulphonic acid); H/L, heterophil to lymphocyte ratio; ↑, increase; ↓, decrease.

**Table 8 toxins-13-00217-t008:** Frequencies of effects of the nonspecific biomarkers of toxicity of deoxynivalenol (DON) in chickens at starter and grower cycles of growth.

Biomarkers Frequently Determined ^1^	No Effect		↑, Increase		↓, Decrease	
	Starter (at 21 Days)	Grower (from 21 Days)	Starter (at 21 Days)	Grower (from 21 Days)	Starter (at 21 Days)	Grower (from 21 Days)
**Productive parameters**	[46,48,67,71]	[45,50,57,58,59,60]	[68]	[31,47,52,54,68,70]	[33,49,61,63,64,65,66,69]	[19,27,51,53,55,56,62]
**RW of organs**						
Liver	[63,64,65,66,67]	[19,33,45,53,62,70,76]		[61]		[55,68]
Kidneys	[63,64,65,66,67]	[19,45,53,62,68,70]				[76]
Gizzard	[49,67]	[33,45,62,77]	[63,64,65,66]	[19,68,70,76]		
Small intestine		[33,45,62]			[49]	[19,77]
Bursa of Fabricius	[65,67]	[19,27,33,53,62,68,70,76]	[63,66]			
Spleen	[49,64,65,66]	[19,27,33,53,68,70,76]		[54,61]		[62]
Thymus		[27,33,61,76]		[19]		
Heart	[49,67]	[19,33,45,61,68,70,76]	[63]	[62]		
**Intestinal morphology**						
Villus height		[19,54]	[69]		[49,51,81,82]	[33,50,60,77,80,83]
Crypt depth	[69]	[19,83]	[82]		[51,81]	
**Blood biochemistry**						
Total protein		[62]				[27,88,89,90]
Albumin						[52,88]
ALT	[67]			[88,89,90]		[58]
ALP				[88]		
AST	[67]		[63]	[88]		
Triglycerides				[58]	[64]	[27,68,89,90]
Glucose	[67]	[62]			[65]	[68]
**Blood hematology**						
Hematocrit	[92]	[62]			[64]	[68,92]
Hemoglobin	[67,92]	[62]			[64]	[52,68,70,91,92]
Erythrocytes	[92]				[64]	[52,91,92]
Lymphocytes						[27,53,54]
**Response to common vaccines**						
NDV		[91,92]	[61]	[61]		[62,92,104]
IBV		[52,91]				[27,58,61]
**Cytokines**						
IL-6				[91,102,104]		[103]
IL-8		[106]		[91]		[102]
IL-1β		[50,103]		[91,104]		[106]
IFN- γ		[103]		[91,105]		[104,106]
**Oxidative stress parameters**						
MDA	[121]		[120]	[83]		
TBARS		[122]	[123]	[76,83]		
**Physiological stress parameters**						
Stress index (H/L ratio)		[62,91]		[27,57,69]		

^1^ ALT, alanine transaminase; ALP, alkaline phosphatase; AST, aspartate transaminase; NDV, Newcastle disease virus; IBV, Infectious bronquitis virus; IL-6, interleukin 6; IL-8, interleukin 8; IFN-γ, interferon gamma; IL-1β, interleukin 1-β; MDA, malondialdehyde; TBARS, thiobarbituric acid reactive substances; H/L, heterophil to lymphocyte ratio.

## Data Availability

Not applicable.

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
