# Peer review of "Biomarkers of Deoxynivalenol Toxicity in Chickens with Special Emphasis on Metabolic and Welfare Parameters"

_toxins, 2021, doi:10.3390/toxins13030217_

Round 1
Reviewer 1 Report
OVERALL EVALUATION.
The manuscript cover an important subject but, in my opinion, although this is a descriptive review and not a meta-analysis, Authors should better investigate the relationship between DON intake and biological effects.
Therefore manuscript can be accepted only after major revision. The main shortcomings of the paper are listed below.
Line 98. Could be useful to specify the guidance level of DON
Line 99. There is a contradiction with the previous statement “the concentration of DON in plasma was un-96 der the limit of detection (LOD) which ranged from 2 to 7 ng/mL, when broilers were fed 97 lower or close to the guidance level of DON in poultry feed”. This is probably due to the LOD higher than LOQ. Authors need to better explain this aspect.
Line 241. The absence of effect could be due to a level of DON contamination sharply lower than ones adopted in the research cited in lines 233—235.
Tables 2, 3,4 and 5. Since Authors state that length of the experiment affect the response of animals to DON intake (and I agree), could be useful to group the data according to the more frequent length of the experiment (e.g. 21 and 35 days) and see if, for the same duration, there is a clear relationship between DON contamination and biological parameters.
Author Response
We would to thank you for taking the time to assess our manuscript and we have addressed all the comments you raised.
"Please see the attachment".

Reviewer 2 Report
The manuscript entitled “Biomarkers of Deoxynivalenol Toxicity in Chickens with Special Emphasis on Metabolic and Welfare Parameters” is is an interesting topic and the manuscript is very well written. I congratulate the authors, but I would like to suggest some improvements.

Author Response
We would to thank you for taking the time to assess our manuscript and we have addressed all the comments you raised.
Please see the attachment.

Reviewer 3 Report
Abstract
L5: „most widespread mycotoxin in poultry feed“ – worldwide? Or in certain countries? Can web e sure that this is not oversimplified – maybe there are mycotoxins which are also very widespread but not measured so frequently?
Introduction
L 18: Please give a reference – is it only the increasing world population or also an increased demand for meat per person (especially in highlydeveloped countries) leading tot he increase in poultry meat consumption?
L23-L26 Please provide references.
L30: „As DON is the most frequently encountered mycotoxin in cereal commodities [2], it 30 is not surprising that DON is a major contaminant of poultry feed.“ – Is this a worldwide problem or just in certain regions? Please provide detailed information!
L43: I think „vomiting“ refers to pigs rather to chickens. To use the word „vomiting“ in chickens having a crop plus gizzard and ventriculus would anatomically not be appropriate. I would rather call this regurgiation or reflux? But the question in this case remains if the clinical sign of „vomiting“ is even present in birds! Please recheck! However acute DON-intoxication causes oral and crop necroses in chicken.
L56: „and this is the most common practice used today“ – please give numbers here – how many percent of poultry flocks use these detoxifiers? Are there regional differences?
L 58: I would rephrase this: „To prove the efficacy and safety of these feed additives“ – otherwise the reader gets the impression that these detoxifiers are not properly tested – but they are on the market. This could sound confusing…
L62: change to „research has been conducted“
Biomarkers of exposure
L128: Can the inconsistent results in these studies be partially be explained by different levels of 3 and 15- A-DON in the feed? Was it measured in feed in the previously mentioned studies?
L 152: Please correct to „species“
L 155: Is there evidence that the crop may play a role in the detoxification of DON as it does so in many other avian species concerning other toxic products!
L 173/174: „the toxicokinetic behavior of DON in chickens was characterized by low absorption“ this statement contradicts with L 160 „Then, they suggested that after absorption, DON is extensively biotransformed“. The question remains how much of the DON is actually absorbed as DON. Please propose a metabolism pathway to DON-3S. If not enough information is available to do so, please give some detailed indications which aspects regarding this topic future research should consider. Is there a potential role of the microbiome (crop?) in the detoxification process? What role might the kidney play (birds have a renal portal system).
Biomarkers of effect
3.1. Productive parameters
Please add information to whether the contamination of DON in the feed in different studies was a natural contamination or if only DON was added to the diet (table). The presence of the fungus in grains might have an effect on the digestibility and therefore increase digestibility (see Daenickes literature) which might increase peformance.
Please add also the broiler breed (Cobb, Ross, etc to the table).
Moreover, please consider improvement of genetics of broiler chickens over the years in the discussion of growth effects. Maybe rapidly growing birds are more susceptible to detrimental effects on growth?
3.2 Relative weight of organs
L 318. Please discuss the reasons for the contradiction of these results
Please give some hints to the veterinarians in the field which parameters are most suitable
L 332: rephrase „intestinal density“ in this context.
3.3. Intestinal morphology
L367-70: Please add the biomarker-context to your conclusion. Can the veterinarian in the field use morphological parameters of the intestine? Overall this part of the manuscript nicely
summarize previous literature on the topic BUT I am currently missing new thoughts/ ideas of the authors how to apply this knowledge for practice or future research! This should be improved!
3.4. Biochemical and hematological parameters
Table 5: Please indicate if the reported effects are just statistically significant or also biologically significant. Are the values reported in the reviewed studies still considered physiological or already pathologic?
L 430: Please give a statement on which parameters are the most promising and in which direction future research should go?
3.5.1 Biomarkers related to humoral immune response
L 470: Regarding titers for vaccine: broiler chickens are normally full of maternal antibodies. This needs to be considered conducting research involving vaccinations as broiler chickens will not mount their own immune response when they still have enough maternal antibodies neutralizing the vaccine. Due to their short life-span in practice, they are only vaccinated against IB and IBD if there is a threat of infection or for IBD if the parent flock was already old (means less maternal antibodies. ND-vaccination however is obligatory in some countries.
Therefore I think the vaccination scheme of parent birds and the maternal antibody titers of chicks used in studies as reported should be mentioned in future literature.
3.5.2.
L 509 This conclusion is a bit weak… Please give a sound outlook and more details here.
3.5.3.
L 532, 535, 547: Please note that the TNF-alpha in chickens was only discovered very recently (see Rohde et al. 2018: https://www.frontiersin.org/articles/10.3389/fimmu.2018.00605/full ). Please recheck (e.g. blast in NCBI) the primers of the studies cited here to see if TNF-alpha was measured here or something else…
Table 6: Did studies investigating cytokines in plasma do this with ELISAs? If yes: where they specific for chickens? Please adress this issue in the chapter!
L586: rephrase „Dänicke“
Conclusions
L 626-628 – please give more detailed information where the focus of future investigations should be! Maybe we should also consider that a lot of the research cited in this review is not conducted in field conditions. Especially regarding biomarkers, I think it would be necessary to increase the proportion of research undertaken in field conditions!
Author Response
First, we have to thank the reviewer for these very constructive comments which helped as to interpret better the information’s along this review.
Please, see the file attached

Round 2
Reviewer 1 Report
OK, you have improved your manuscript that, in my opinion, can be appepted.
Author Response
Dear reviewer,
We would like to thank you for taking the time and effort necessary to revise the revised version of our manuscript.
Reviewer 3 Report
Dear authors,
The manuscript improved considerably. Please find my comments below:
- L99-101 Please reword to make the sentence easier to follow.
- L327 Please correct to “performance”
- “ Response 20a: For the reference: Ghareeb et al., 2013, they mentioned that is TNF-alpha with the corresponding primer consequences CCCCTACCCTGTCCCACAA TGAGTACTGCGGAGGGTTCAT Which is the same mentionned by Lu et al., 2018 F: CCCCTACCCTGTCCCACAA R: TGAGTACTGCGGAGGGTTCAT”
=> Blasting the respective primer sequence for Gallus gallus in NCBI (https://www.ncbi.nlm.nih.gov/tools/primer-blast/index.cgi) results in “lipopolysaccharide induced TNF factor” as target. According to Hong et al 2006 (doi 10.1016/j.dci.2005.12.007) this is a transcription factor regulation TNF-alpha expression. I strongly recommend the authors to recheck this primer and adjust the section in the manuscript accordingly.
